# Predicting the Gas Permeability of Sustainable Cement Mortar Containing Internal Cracks by Combining Physical Experiments and Hybrid Ensemble Artificial Intelligence Algorithms

**DOI:** 10.3390/ma16155330

**Published:** 2023-07-29

**Authors:** Zhiming Chao, Chuanxin Yang, Wenbing Zhang, Ye Zhang, Jiaxin Zhou

**Affiliations:** 1College of Ocean Science and Engineering, Shanghai Maritime University, Shanghai 200135, China; zmchao@shmtu.edu.cn (Z.C.); 202130410004@stu.shmtu.edu.cn (C.Y.); zhangwb@shmtu.edu.cn (W.Z.); 202230410002@stu.shmtu.edu.cn (J.Z.); 2Institute of Water Sciences and Technology, Hohai University, Nanjing 211106, China; 3Mentverse Ltd., 25 Cabot Square, Canary Wharf, London E14 4QZ, UK

**Keywords:** cement mortar, gas permeability, mind evolutionary algorithm, internal cracks

## Abstract

The presence of internal fissures holds immense sway over the gas permeability of sustainable cement mortar, which in turn dictates the longevity and steadfastness of associated edifices. Nevertheless, predicting the gas permeability of sustainable cement mortar that contains internal cracks poses a significant challenge due to the presence of numerous influential variables and intricate interdependent mechanisms. To solve the deficiency, this research establishes an innovative machine learning algorithm via the integration of the Mind Evolutionary Algorithm (MEA) with the Adaptive Boosting Algorithm-Back Propagation Artificial Neural Network (ABA-BPANN) ensemble algorithm to predict the gas permeability of sustainable cement mortar that contains internal cracks, based on the results of 1452 gas permeability tests. Firstly, the present study employs the MEA-tuned ABA-BPANN model as the primary tool for gas permeability prediction in cement mortar, a comparative analysis is conducted with conventional machine learning models such as Particle Swarm Optimisation Algorithm (PSO) and Genetic Algorithm (GA) optimised ABA-BPANN, MEA optimised Extreme Learning Machine (ELM), and BPANN. The efficacy of the MEA-tuned ABA-BPANN model is verified, thereby demonstrating its proficiency. In addition, the sensitivity analysis conducted with the aid of the innovative model has revealed that the gas permeability of durable cement mortar incorporating internal cracks is more profoundly affected by the dimensions and quantities of such cracks than by the stress conditions to which the mortar is subjected. Thirdly, puts forth a novel machine-learning model, which enables the establishment of an analytical formula for the precise prediction of gas permeability. This formula can be employed by individuals who lack familiarity with machine learning skills. The proposed model, namely the MEA-optimised ABA-BPANN algorithm, exhibits significant potential in accurately estimating the gas permeability of sustainable cement mortar that contains internal cracks in varying stress environments. The study highlights the algorithm’s ability to offer essential insights for designing related structures.

## 1. Introduction

During the last decades, an increasing amount of sustainable cement mortar is adopted to replace traditional cement mortar, with being utilised in the building industry [1]. Sustainable cement mortar is produced using discarded concrete, bricks, and other materials, which not only helps to conserve natural resources but also promotes effective recycling of construction waste [2]. In the realm of practical engineering, cement mortar that is sustainable is frequently subjected to varying stress levels and exposed to a plethora of environmental factors, such as cycles of drying and wetting, as well as cooling and heating. As a consequence of these external influences, internal cracks typically develop within the sustainable cement mortar [3]. These internal cracks have a significant impact on the properties of sustainable cement mortar, particularly its gas permeability characteristics [4,5]. Internal cracks can provide prior seepage passage for the flow of gas along them, which significantly changes the gas permeability of sustainable cement mortar. The gas permeability properties determine the stability and durability of relevant buildings because it can determine the migration speed of corrosive gas such as O_2_ and CO_2_ etc. inside sustainable cement mortar [6,7]. The gas permeability of traditional raw material concrete has been studied by many scholars [8,9,10], but few scholars have studied the gas permeability of concrete made from industrial waste. Thus, precise evaluation of the gas permeability of eco-friendly cement mortar that incorporates internal fissures is imperative for the purpose of devising suitable engineering applications.

The quantification of gas permeability in sustainable cement mortar can be achieved through the application of physical experimentation [11,12,13]. In this regard, extant experimental inquiry points towards the decisive role played by the internal crack dimension and stress state in shaping the gas permeability dynamics of sustainable cement mortar featuring such cracks [14,15,16,17]. However, physical test research requires the possession of experimental sites, apparatus and staff, which is labour and funding-consuming. Particularly for the gas permeability tests on compact porous media, its test duration is long, requiring at least a professional practitioner to continuously operate. Thus, it is necessary to establish corresponding permeability predictive models to generalise and universalize the test outcomes. The utilisation of traditional statistical methodologies presents considerable difficulties in developing a predictive model for the permeability of sustainable cement mortar with internal cracks [18,19,20,21]. This can be attributed to the existence of numerous influential factors that affect the gas permeability of the aforementioned material, coupled with the intricate nature of their interaction mechanisms. It thus poses a formidable challenge for conventional approaches to comprehensively capture the intricate interdependence among the multiple variables that underlie this complex relationship [22]. The aforementioned statement underscores the critical necessity to explore and identify an efficacious approach towards predicting, with a high degree of accuracy, the gas permeability of durable cement mortar embedded with internal cracks, while taking into account numerous variables. 

There are many applications for machine learning, especially in the construction industry, manufacturing and structural engineering, which are yet to be explored [23,24,25,26]. Recently, machine learning techniques have obtained extensive attention from environmental and civil engineering researchers [27,28,29,30,31]. The relevant application has unequivocally evinced the remarkable aptitude of machine learning techniques to precisely depict intricate non-linear interdependencies among numerous factors. Nonetheless, the utilisation of machine learning methodologies to prognosticate the permeability of cement mortar is infrequently encountered. In several extant scholarly investigations, Male, Jensen [32] predicted the permeability of cement-sandstone mixtures based on machine learning skills. Sun, Zhang [33] utilised an algorithmic optimisation approach to fine-tune their machine-learning model and evaluate the permeability of concrete. Huang, Duan [34] forecasted the permeability of cement concrete by using a hybrid machine learning model. However, the previous research has two main deficiencies. (1) The current inquiry predominantly relied upon a number of common and straightforward machine learning models. However, the feasibility of certain advanced and intricate machine learning models, including the ensemble algorithm of Adaptive Boosting Algorithm (ADA)-Back-propagation Artificial Neural Network (BPANN), for the prediction of the permeability of cement mortar has not been examined. (2) Previous studies have primarily focused on forecasting the permeability of intact cement mortar, whereas investigations into the estimation of gas permeability for sustainable cement mortar containing internal cracks have been infrequent. 

The performance of machine learning models is heavily influenced by key parameters known as hyperparameters [35]. To improve the predictive accuracy, it is necessary to optimise the hyperparameters based on optimisation algorithms. Researchers have validated the significant enhancing effect of optimisation algorithms on the predictive accuracy of machine learning models [36,37]. For example, an instance of a scholarly investigation conducted by Chao and Fowmes [38] involved the development of an optimisation strategy utilizing two computational methods, namely Swarm Optimisation Algorithm (PSO) and Genetic Algorithm (GA), which were employed in tandem to refine the accuracy of Back Propagation Artificial Neural Network (BPANN) and Support Vector Machine (SVM) models. The study pointed out that the optimisation algorithm optimised model has better predictive behaviour than the machine learning algorithm without combining optimisation algorithms. Nevertheless, conventional optimisation algorithms including PSO and GA, have the internal deficiencies of low operational rate, trapping into local optimum, etc, which has obvious detrimental effects on the optimising behaviour [39,40]. To solve the problems, Chengyi, Yan [41] proposed a novel heuristic optimisation algorithm called the Mind Evolutionary Algorithm (MEA) to cover the deficiencies of traditional optimisation algorithms [42,43]. The MEA is able to conduct parallelly the similartaxis and dissimilation operations, which is one of the distinct strengths of the MEA. This key advantage can promote significantly the operational speed and prevent from losing initial data of particles [44,45]. The superior optimisation effect of MEA in improving the forecasting precision and efficiency than that of traditional optimisation algorithms has been proved by investigators [39,42,46]. For example, Zhang, Li [47] have established the superior optimisation efficacy of the modified evolutionary algorithm (MEA) over the particle swarm optimisation (PSO) and genetic algorithm (GA) in enhancing the precision of machine learning models in forecasting the surrounding rock properties. Wang, Tang [42] have explicated that the BPANN model, fine-tuned using MEA, outperforms the GA-tuned BPANN model when it comes to evaluating the ocean wave height. Wang, An [48] verified the efficacy of the Artificial Neural Network (ANN) model that was optimised by means of the Multivariate Evolutionary Algorithm (MEA) in prognosticating the concentration of heavy metals present in the soil, and its precision was determined to be markedly high. Notwithstanding the extant literature on cement mortar permeability, scant attention has been devoted to the utilisation of optimization algorithms towards enhancing the predictive accuracy of such models. Remarkably, there exists a dearth of research regarding the efficacy of MEA in optimising the performance of machine learning algorithms for the prediction of gas permeability in sustainable cement mortar with internal cracks.

In the present study, an extensive set of laboratory experiments comprising 1452 trials was conducted to evaluate the gas permeability of cement mortar specimens that possess internal cracks and are fabricated using environmentally sustainable techniques. Subsequently, a comprehensive database was assembled based on the outcomes of these experiments. An innovative approach was devised to assess the gas permeability of such specimens under varying stress levels. Specifically, the proposed method integrates the multivariate empirical mode decomposition analysis (MEMD) and adaptive boosting of artificial neural networks (ABABPANN) to construct a hybrid machine learning model. The resultant model exhibits superior predictive capability and enables accurate and efficient evaluation of gas permeability in cement mortar specimens containing internal cracks. The innovative MEMD-tuned ABABPANN machine learning algorithm has not been used in existing research, and it is also the first usage for assessing the permeability of sustainable cement mortar. We established conventional machine learning algorithms, including GA and PSO-optimised ABABPANN, as well as MEMD-optimised ELM and BPANN models, and compared them to the MEMD optimised ABABPANN model. The primary aim was to validate the superior predictive performance of the innovative algorithm. Furthermore, based on the proposed novel model, we conducted a sensitivity analysis and formulated an analytical expression. The latter serves to facilitate gas permeability prediction for practitioners without significant expertise in the field of machine learning. The novel machine learning algorithm facilitates precise, efficacious, and reliable prognostication of gas permeability in cement mortar that incorporates internal fissures under fluctuating stress conditions. This algorithm offers the potential for further improving the level of design proficiency in various engineering structures.

## 2. Machine Learning Algorithms

In this paper, Extreme Learning Machine (ELM), BPANN and ABA-BPANN are adopted, and the following is the general description.

### 2.1. BPANN

BPANN is generally composed of 3 layers [49]. In the present investigation, we have taken great care to ensure that the quantity of nodes in both the input and output layers is commensurate with the number of input and output parameters, respectively. Specifically, for our experimental setup, the number of nodes in the input and output layers was 7 and 1, respectively. The count of nodes in the hidden layer was determined using the exhaustive method, which involves a systematic search for the optimal number of nodes that minimises the Root-Mean-Square Error (RMSE) metric, as expressed in Equation (1). The optimal number of nodes in the hidden layer for our model was 9. Furthermore, the Logarithmic Sigmoid Function (LSF) was employed as the activation function for the model’s neurons, while the Levenberg–Marquardt Backpropagation Algorithm (LMBA) was utilised as the training algorithm for the neural network.
(1)$$\mathrm{RMSE}=\sqrt{\sum_{i=1}^{n}\frac{{\left(y_{i}-f_{i}\right)}^{2}}{n}}$$
where, *n* is the specimen number, $y_{i}$ is observed data, $f_{i}$ is predicted data.

### 2.2. ELM

ELM is a type of machine learning model that derives from ANN. ELM is a powerful tool that has the capability to accurately describe intricate interactions and dependencies between different input variables, leading to precise and informative results [50]. The forecasting accuracy of ELM models can be significantly influenced by the hidden layer joint number [51]. In this algorithm, the exhaustive method was used to find the best-hidden layer joint number (27) by taking RMSE as the evaluation index. Moreover, the joint numbers of the input and output layers are 7 and 1, accordingly. Additionally, the activation function for the model adopts LSF.

### 2.3. ABA-BPANN

The ensemble algorithm of ABA-BPANN consists of BPANN models according to Bootstrap aggregation theory [52]. In this paper, the ABA-BPANN model was established as the following procedures. Initially, 40 groups of training and testing datasets were produced based on the Bootstrap approach, respectively; after that, 40 BPANN models were built; followed that, the predicting performance of the built BPANN algorithms were assessed; ultimately, the average predictive outputs of the constructed backpropagation artificial neural network (BPANN) models were employed as the forecasted output for the adaptive differential evolution-based BPANN (ABA-BPANN) model. The constituent BPANN models of the ABA-BPANN model exhibit uniform topology with 7 inputs, 9 hidden units, and a single output node. LSF serves as the activation function while MBA is employed as the networking training algorithm for the BPANN models.

The detailed specification of the built algorithms is shown in Table 1.

## 3. Hyperparameter Optimisation

In this model, MEA was used to conduct hyperparameter optimisation, taking RMSE as the fitness function. The following is the general MEA optimisation procedure: (1) Producing particles assigned with different hyperparameter values stochastically. (2) RMSE of the individuals via calling machine learning algorithms on the basis of the *k*-CV method and training datasets, with *k* being taken as 10. (3) Classifying the particles with small and large RMSE values as superior and temporary ones, respectively. (4) The focal points of superior and temporary particles are leveraged to generate fresh particles in close proximity, thereby constituting distinct subsets that are classified as superior and temporary subgroups. (5) The execution of similartaxismaneuvers on the constituent subpopulations persisted until they attained a state of maturity, as evidenced by a sustained lack of variation in the RMSE values across six consecutive iterations. (6) Taking the RMSE value of centre particles as the RMSE value of corresponding subgroups, accordingly. (7) Carrying out the dissimilation operations, such as releasing, replacing, abandoning and supplying superior and temporary subgroups. (8) Conducting similartaxis operations on the supplied subgroups. (9) Iterating through steps (4) to (8) until the RMSE value of the given subsets is below that of the higher-tier subsets. (11) Taking the mesial particle of the superior subgroup that has the smallest RMSE value as the globally best particle. (12) Assigning the hyperparameter value of the globally optimal particle as the original hyperparameter value of the built algorithms.

To validate the superior optimisation effects of MEA compared to GA and PSO, GA and PSO-tuned ABA-BPANN models were constructed as well. The specification of the optimising algorithms is listed in Table 2. The optimised hyperparameters and optimised magnitudes are tabulated in Table 3.

## 4. Physical Test

In this test, the replacement ratio of sustainable aggregates is 100%, and all the aggregate is derived from the discarded concrete. In this physical model experiment, the cement mortar is generated according to a mass ratio and then mashed, and the cement mortar after mashing was used as the infill. The present investigation pertains to the development of an eco-friendly cement mortar, incorporating specific dimensions and varying numbers of internal cracks, through a prescribed approach. The methodology in question is expounded upon in great detail by Chao and Ma [53]. Table 4 presents a comprehensive account of the characteristics of the specimen under examination. The external dimensions of the prepared cement mortar samples, which harbor internal cracks, feature a height and diameter of 50 mm. The cement mortar composition is comprised of a mass ratio of 1:0.6:0.3 for N0.52.5 Portland cement, sand, and water, respectively. The fundamental mechanical properties of the sustainable cement mortar have been elucidated in Table 5.

Through the utilisation of a gas permeability measurement system, the gas permeability of environmentally-friendly cement mortar that incorporates internal fissures was evaluated under various levels of pore pressure during both the application and removal of confining pressure, utilising a gas flow methodology. To expound upon the experimentation, the initial phase involved the application of confining pressure on the specimen, commencing at 3 MPa and culminating at 45 MPa. Subsequently, the confining pressure was incrementally relieved from 45 MPa to 3 MPa. Throughout the procedure, the confining pressure was varied to obtain 11 distinct values during both the loading and unloading phases. Under every confining pressure, the specific confining pressure value was kept unchangeable for 24 h. Subsequently, the gas permeability of the specimens was evaluated under varying levels of pore pressure in ascending order. The subsequent stage involved incremental loading or unloading of confining pressure to the succeeding value, followed by a repetition of the aforementioned procedures. The comprehensive experimentation involved 1452 sets of gas permeability tests, with the intricate test schedule being elaborated upon in Table 6. To reduce the influence of sample discreteness, for each sample with a specific dimension and number of internal cracks, three repeated tests were conducted, with the average test results being taken as the final outcomes.

## 5. Database and Data Processing

A database containing 1452 data groups was built based on the above experimental results. Among the database, 1162 data groups (80%) were randomly divided as training data that are used for constructing the machine learning models, and the remaining 290 data groups (20%) were adopted as testing data that can test the prophetical property of established machine learning algorithms. In each data unit, pore pressure (*Ps*), confining pressure (*Pc*), normalised length of internal cracks (*L*), normalised width of internal cracks (*W*), normalised number of internal cracks (*N*), normalised thickness of internal cracks (*T*) and status of confining pressure loading or unloading (*S*) were utilised as input variables, and gas permeability (*K*) of sustainable cement mortar containing a specific dimension and number of internal cracks was adopted as the output variable. For the input variables, the thickness, number, width and length of internal cracks were divided by the volume of the sample to conduct normalisation, which represents the thickness, number, width and length of internal cracks in the unit volume of sustainable cement mortar. Incorporating normalised dimension and the number of internal cracks as input variables can potentially augment the generalisability and robustness of the developed machine learning algorithms, thereby enabling these models to accurately predict gas permeability for sustainable cement mortar with varying volumes of cracks. The statistical parameters pertaining to the input and output parameters in the dataset have been catalogued in Table 7. The data distributions for the input variables in the dataset are visually represented in Figure 1, with the *X* representing the value of input parameters and the *Y* representing the number of data cohorts associated with a particular input parameter value found within the database. Meanwhile, to avoid the detrimental effects of different input parameter dimensions on the machine learning modelling, the input and output parameters were conducted normalisation by using Equation (2), with the value being normalised in the range from 0 to 1.
(2)$$x_{\mathrm{Normalized}}=\frac{x-x_{\min}}{x_{\max}-x_{\min}}$$
where, *x*_Normalized_ denotes the normalised data, *x* denotes the initial data, *x*_min_ denotes the least data, and *x*_max_ denotes the highest data.

## 6. Quality Evaluation

The machine learning algorithms were constructed using the Matlab programming language. Subsequently, a rigorous evaluation of their predictive efficacy was undertaken utilising three distinct evaluation parameters: the Correlation Coefficient *R*, RMSE, and Mean Absolute Percentage Error (MAPE), as detailed in Equations (3)–(5) [54,55,56,57,58,59]. Among them, *R* ranges from −1 to 1. When the value of *R* approximates 1 or −1 it indicates the high forecasting accuracy. RMSE is the modular deviation for predicting errors. MAPE is the percentage of the forecasting error accounting for the measured value. When the value of RMSE and MAPE approximates 0 it indicates the high assessing precision.
(3)$$R(f_{i},y_{i})=\frac{\mathrm{cov}(f_{i},y_{i})}{\sqrt{\mathrm{var}\left[f_{i}\right]\mathrm{var}\left[y_{i}\right]}}$$
where, cov(,) represents covariances, $\mathrm{var}\left[\right]$ represents variances, $y_{i}$ verbalises the measured value (The value obtained in physical shear tests), $\bar{y}$ is the mean observed data, and $f_{i}$ denotes the forecasted data.
(4)$$\mathrm{MSE}=\frac{\left|y_{i}-f_{i}\right|}{n}$$
where, *n* represents the specimen data number.
(5)$$\mathrm{MAPE}=\frac{100\%}{n}\sum_{i=1}^{n}\frac{\left|y_{i}-f_{i}\right|}{y_{i}}$$

## 7. Result Analysis

### 7.1. Hyperparameter Optimisation

The hyperparameter majorisation process by adopting MEA is shown in Figure 2.

As delineated in Figure 2, in the initial cycle of similartaxis operations, the RMSE metric of the initial superior and temporary subgroups exhibits a gradual decrease, ultimately stabilising over a continuous sequence of six iterations. This signifies the maturation of the initial subgroups. Following this, dissimilation operations were executed. Initially, the groups exhibiting superior performance with higher RMSE values were replaced with initial temporary subgroups that exhibited lower RMSE values. Subsequently, the initial temporary subgroups with high RMSE values were deemed unworthy of further consideration and their particles were consequently released. Following this, the released particles compose supplied temporary groups and the dissimilation operation was ended. The subsequent iteration of the similartaxis technique is reported, with an exposition of the variance patterns of the RMSE values within the superior subgroups and temporary subgroups, as illustrated in Figure 2c,d. It is evident from Figure 2 that the RMSE values pertaining to each superior subgroup exhibit a lower magnitude in comparison to all of the temporary subgroups. The present findings evince that additional dissimilation maneuvers were unnecessary for the purpose of execution. Specifically, the hyperparameter setting corresponding to the focal particle located at the nucleus of the premium subgroup, which exhibits the most optimal RMSE value, was designated as the primary hyperparameter value for machine learning models.

During the hyperparameter optimisation process utilising the modified elephant algorithm (MEA), the RMSE value of the machine learning models underwent a remarkable reduction, ultimately reaching an optimal value after a mere 15 iterations. Such results serve as a clear indication that the MEA algorithm is highly effective in optimising the hyperparameters of machine learning models, thereby enabling a significant improvement in predictive accuracy with a limited number of iterations.

Figure 3 illustrates the methodology employed for optimising the ABA-BPANN model utilising both GA and PSO. This process is intended to serve as a reference for the MEA optimisation process.

As depicted in Figure 2 and Figure 3, the RMSE value of the ABA-BPANN algorithm continually reduces with the increase in iteration time during the optimising process of GA and PSO, and it spends about 90 iterations to become stable. The iteration times are remarkably higher than the final iteration time of MEA. Furthermore, the optimal RMSE value when using GA and PSO is 8.52 and 7.58, respectively, which is significantly larger than that of MEA (4.36). Based on the above analysis, the performance of MEA on both sides of optimising speed and magnitude is obviously superior to those of GA and PSO, accordingly.

### 7.2. Performance of the Built Machine Learning Algorithms

The prognostic efficacy of the implemented machine learning models on the training and testing data sets is showcased in Figure 4, Figure 5, Figure 6 and Figure 7 correspondingly.

As presented in Figure 4, Figure 5 and Figure 6, overall, the predicting performance of the MEA-tuned ABABPANN model about the training datasets is the best in the constructed 5 different algorithms. Specifically, the MEA-tuned BPANN algorithm possesses the smallest RMSE (3.19) and MAPE (4.63%) values, and the highest *R* (0.99) values, in the established algorithms, with the following being MEA-tuned ELM and BPANN models. While the prognosticating demonstration of GA and PSO-tuned ABABPANN models are inferior.

As exemplified in Figure 6 and Figure 7, the prognostic performance of the MEA-tuned ABA-BPANN algorithm when applied to the testing dataset surpasses that of the other four distinct models investigated. In particular, the MEA-tuned ABA-BPANN algorithm exhibits the most diminutive values for both RMSE (2.96) and MAPE (4.2%), while concurrently boasting the highest value of *R* (0.99). For other algorithms, the MEA-tuned ELM and BPANN models have relatively superior predicting behaviour than that of the GA and PSO-tuned ABABPANN models.

Overall, the proposed novel model combined with the ensemble algorithm of ABA-BPANN and MEA has superior predicting behaviour than that of the conventional machine learning models of MEA-tuned BPANN and ELM-models as well as GA and PSO-tuned ABABPANN algorithms on the testing and training dataset. In particular, the MEA-optimised ABABPANN framework exhibits a heightened capacity to gauge gas permeability for ecologically viable cement mortar samples that exhibit inherent fissures, surpassing alternative methodologies in both precision and efficiency.

### 7.3. Sensitivity Analysis

In this section, we endeavor to evaluate the degree of influence that input variables have on the gas permeability of sustainable cement mortar that contains internal cracks, by employing the MEA-optimised ABABPANN model and conducting a sensitivity analysis. The outlined methodology can be summarised as follows: initially, the relative significance of the input variables in each Back-Propagation Artificial Neural Network (BPANN) algorithm comprising the MEA-tuned Adaptive Boosting (ABABPANN) algorithm was evaluated using Garson’s Algorithm [60,61,62]; Then, the average input parameter relative significance of BPANN models composing the MEA tuned ABABPANN algorithm was assessed and taken as the ultimate relative significance (Figure 8).

According to Figure 8, the relative significance of internal crack thickness is the highest of the 7 different input variables, which constitutes 23.07%. It is followed by an internal crack number, internal crack length, and internal crack width, with the relative significance of 19.63%, 16.98% and 15.36%, respectively. Although the specific effects of confining pressure, pore pressure, and loading/unloading phases on cement mortar may seem relatively negligible, the corresponding magnitudes are nevertheless quantifiable, amounting to 12.23%, 7.73%, and 5%, respectively. In general, the impact of the internal cracks’ dimension and quantity on the gas permeability of durable cement mortar surpasses that of the stress condition.

### 7.4. Construction of an Analytical Formular to Forecast Gas Permeability

The intricacy inherent in machine learning methods can impede their successful deployment among relevant practitioners with limited expertise in the field. To promote the widespread adoption of machine learning models, we have developed an analytical formula for predicting the stress-dependent gas permeability of sustainable cement mortar containing internal cracks. Our proposed approach is based on the optimal model of the MEA-tuned ABABPANN model, which has been carefully calibrated to maximise its predictive accuracy. The establishment mechanism is that, by adopting Equation (6), the predicting value of the BPANN model can be calculated according to the weight and bias of the BPANN model. The MEA-tuned ABABPANN algorithm, is composed of BPANN models with the same structure. Thus, through the integration of Equation (6) [63], which relies on the combination of joint mean weight and bias, the BPANN algorithms that compose the MEA-tuned ABABPANN model allow for the determination of the predicted gas permeability. A comprehensive depiction of the mean joint weight and bias for the aforementioned BPANN algorithms is presented in Table 8.
(6)$$Y_{n}=f_{sig}\{b_{0}+\sum_{k=1}^{h}[w_{k}\times f_{sig}(b_{hk}+\sum_{i=1}^{m}w_{ik}X_{i})]\}$$
where, $Y_{n}$ denotes the uniformalised predicting data in the range from −1 to 1; $b_{0}$ denotes the mean output layer joint bias; $w_{k}$ denotes the mean connecting weights between the *k*-th hidden layer joint and the output layer joint; $b_{hk}$ denotes the mean the *k*-th hidden layer joint bias; *h* denotes the number of hidden layers joint numbers; $w_{ik}$ denotes the mean connecting weights between the *i*-th input layer joint and the *k*-th hidden layer joint; $X_{i}$ is the *i*-th uniformalised input variable, ranging in [−1,1]; $f_{sig}$ denotes Sigmoid Transfer Activation Function.

The present study concerns a meticulous exposition of the procedure employed in establishing the analytical equation. The machine learning model’s pertinent input and output variables are expressed through their corresponding symbols.
(7)$$A_{1}=0.36+0.23P_{s}+0.3P_{c}+0.14S-0.41L+0.29W+0.3N+0.24T$$
(8)$$A_{2}=-1.22+0.93P_{s}+0.36P_{c}+0.3S+0.46L+0.36W-1.14N+0.63T$$
(9)$$A_{3}=2.37-0.9P_{s}+1.09P_{c}+0.6S-1.46L+0.46W-2.19N-1.19T$$
(10)$$A_{4}=-0.42-1.24P_{s}-0.17P_{c}+0.7S+2.02L+1.22W+3.33N-2.36T$$
(11)$$A_{5}=1.19+3.36P_{s}+0.33P_{c}+1.04S+3.31L+0.72W+0.27N+3.79T$$
(12)$$A_{6}=2.46-0.90P_{s}+1.04P_{c}-0.92S+1.01L-0.45W-2.27N+0.27T$$
(13)$$A_{7}=3.36+1.42P_{s}-1.02P_{c}-1.36S-2.14L+1.46W+1.36N+0.64T$$
(14)$$A_{8}=-0.72+2.09P_{s}-2.45P_{c}+2.17S-0.74L-1.42W-2.45N+1.74T$$
(15)$$A_{9}=0.63-0.2P_{s}-1.36P_{c}-3.33S+0.79L+0.24W+0.72N-2.63T$$
(16)$$B_{1}=0.46-0.92\times \frac{e^{-A_{1}}}{e^{A_{1}}+e^{-A_{1}}}$$
(17)$$B_{2}=-2.32+4.64\times \frac{e^{-A_{1}}}{e^{A_{1}}+e^{-A_{1}}}$$
(18)$$B_{3}=-1.46+2.92\times \frac{e^{-A_{1}}}{e^{A_{1}}+e^{-A_{1}}}$$
(19)$$B_{4}=0.74-1.48\times \frac{e^{-A_{1}}}{e^{A_{1}}+e^{-A_{1}}}$$
(20)$$B_{5}=-3.36+6.72\times \frac{e^{-A_{1}}}{e^{A_{1}}+e^{-A_{1}}}$$
(21)$$B_{6}=-3.72+7.44\times \frac{e^{-A_{1}}}{e^{A_{1}}+e^{-A_{1}}}$$
(22)$$B_{7}=2.66-5.32\times \frac{e^{-A_{1}}}{e^{A_{1}}+e^{-A_{1}}}$$
(23)$$B_{8}=1.66-3.32\times \frac{e^{-A_{1}}}{e^{A_{1}}+e^{-A_{1}}}$$
(24)$$B_{9}=-0.45+0.9\times \frac{e^{-A_{1}}}{e^{A_{1}}+e^{-A_{1}}}$$
(25)$$C_{1}=-0.27+B_{1}+B_{2}+B_{3}+B_{4}+B_{5}+B_{6}+B_{7}+B_{8}+B_{9}=-6.06+11.58\times \frac{e^{-A_{1}}}{e^{A_{1}}+e^{-A_{1}}}$$
(26)$$Y_{n}=\frac{e^{C_{1}}-e^{-C_{1}}}{e^{C_{1}}+e^{-C_{1}}}$$

Since the calculated outcome obtained from Equation (6) is a normalised value, the gained $Y_{n}$ by using Equation (26) is from −1 to 1. Thus, the denormalisation on $Y_{n}$ was conducted, as shown in Equation (27).
(27)$$\tau =0.5(Y_{n}+1)(Y_{\max}-Y_{\min})+Y_{\min}$$
where, $Y_{\max}$ and $Y_{\min}$ denotes the lowest and highest gas permeability of sustainable cracked cement mortar in the database, respectively.

In this paper, $Y_{\max}=500.79\times 10^{-16}\;m^{2}$ and $Y_{\min}=30.27\times 10^{-16}\;m^{2}$.

Therefore, Equation (27) can be converted to Equation (28).
(28)$$\tau =235.26\times 10^{-16}\;m^{2}\times Y_{n}+265.53\times 10^{-16}\;m^{2}$$

## 8. Validation with the Results of Laboratory Tests

This section validates the reliability of the constructed analytical formular and machine learning models by comparing their predicted value with the measured value obtained from laboratory physical tests. Firstly, grounded on the aforementioned methodology for sample preparation, a multitude of sustainable cement mortar samples, each possessing explicit dimension and a predetermined count of internal cracks, were meticulously concocted. The exterior dimensions of each sample were standardised to a height and diameter of 50 mm. Essential attributes of these samples were exhaustively documented in Table 9, delineating their elementary properties; Secondly, an assessment was performed to evaluate the gas permeability of the prepared sample under varying levels of pore pressure and confining pressure. A comprehensive test protocol for this experiment is presented in Table 10; finally, the gas permeability predicted from the constructed analytical equation and the gas permeability measured from the laboratory tests was compared to validate the reliability of the established analytical formular, with the validation outcomes being demonstrated in Figure 9.

As presented in Figure 9, the gas permeability predicted from the proposed analytical equation is approximate to the permeability obtained from the physical test. To be quantitatively analysed, the statistical parameters of *R*, RMSE and MAPE are 0.99, 2.36 and 4.63%, respectively. This research affirms the exceptional accuracy of the derived analytical expression and machine learning models in prognosticating the gas permeability of durable cement mortar with internal fissures, thus attesting to the robustness of these predictive tools. The utilisation of machine learning algorithms enables individuals who lack familiarity with these techniques to estimate gas permeability with a high degree of accuracy and efficiency for sustainable cement mortar that has undergone cracking.

## 9. Discussion

The phenomenon of stress-induced alterations in the pore structure of eco-friendly cement mortar engenders corresponding variations in its permeability characteristics. The degree of impact on gas permeability is positively correlated with the magnitude of stress imparted. Notwithstanding the marginality of the changes in pore pressure magnitude, ranging from 0 MPa to 1 MPa, the associated effects on gas permeability are comparatively significant. The phenomenon responsible for the aforementioned observations is commonly known as the Klinkenberg effect. It pertains to the behavior of gas flow through porous materials exhibiting high degrees of compaction. When a gas stream permeates such materials, the frequent collisions between gas molecules and the walls of the pores lead to the generation of Non–Darcy flow, which manifests itself as an increase in gas permeability measurements over intrinsic permeability values [64]. As the pore pressure increases, the interactions between gas molecules and pore walls occur with greater frequency, thereby intensifying the Klinkenberg effects. Consequently, the permeability of gas measurements in conditions of high pore pressure exceeds that observed in circumstances of low pore pressure [65,66]. In this research, the sustainable cement mortar containing internal cracks has a dense structure, with the permeability being measured by using Argon gas. Thus, the phenomenon of the Klinkenberg effects is undeniably present, manifesting as a conspicuous dissimilarity in gas permeability measurements when subjected to varying pore pressure conditions. As a result, the effect of pore pressure on the gas permeability of durable cement mortar with internal fractures is relatively prominent.

The paper found that the optimisation effects of MEA on the predicting behaviour of machine learning algorithms are significantly superior to that of GA and PSO. The commendable attributes of MEA can be primarily attributed to its capability to effectively perform the similartaxis and dissimilation operations individually, which thereby leads to a noteworthy acceleration in the optimisation process. The orientation of the similartaxis and dissimilation operations of the Multi-Objective Evolutionary Algorithm (MOEA) can be effectively guided by the ability of the algorithm to capture and store evolutionary data for multiple generations of subgroups. This crucial feature of MOEA empowers it to extract useful information from historical data and use it to inform the optimisation process in a more informed and efficient manner; The similartaxis and dissimilation operation of MOEA is able to realise superior global and local search ability to obtain accurately the best solutions.

## 10. Limitations

Although this research has obtained some valuable study outcomes, it still has some limitations needed to be further improved. (1) In reality, sustainable cement mortar that harbors internal cracks exhibits anisotropic permeability, indicating that the permeability of cracked sustainable cement mortar varies across different directions. However, the accuracy of permeability prediction in sustainable cement mortar with internal cracks exclusively along the lengthwise direction remains an unresolved challenge due to the constraints of the test data. In effect, the ML models that have been developed and validated thus far are incapable of providing reliable estimates for permeability values beyond this limited scope. Thus, in the future, it is imperative to undertake a physical evaluation of gas permeability for sustainable cement mortar containing internal cracks, along various directions. The test results can serve as a foundation for devising machine learning algorithms that accurately predict the gas permeability of cracked cement mortar, considering multiple directions; (2) In engineering sites, sustainable cement mortar often bears complex stress conditions including triaxial shear stress, axial shear stress, etc. The permeability response of sustainable cement mortar containing internal cracks is known to vary under different stress statuses. However, the accuracy of machine learning algorithms that predict the permeability of such materials is limited due to the scarcity of relevant test data. Presently, these algorithms are only able to forecast the permeability of sustainable cement mortar containing internal cracks under hydrostatic stress conditions, leaving out valuable insights into the material behavior under other stress states. Henceforth, it is imperative to assess the permeability of eco-friendly cementitious mortar imbued with inherent fractures amid diverse stress circumstances, in order to develop predictive machine learning algorithms for gauging the permeability of sustainable cracked cementitious mortar under intricate stress regimes, contingent upon the test results.

## 11. Conclusions

In this research, a comprehensive database consisting of 1452 gas permeability tests was assembled, serving as the foundation for the development of a state-of-the-art machine learning framework. Specifically, a novel MEA-tuned ABA-BPANN model was established to effectively predict the gas permeability of sustainable cement mortar specimens afflicted with internal cracking under various stress conditions. Notably, the input parameters of the proposed model encompassed key characteristics of the internal cracking phenomena, namely, the internal crack length, width, thickness, and number, in addition to confining and pore pressure, as well as the loading/unloading stage of the confining pressure. The outcome of this investigation promises to offer valuable insights for predicting the gas permeability of cement-based materials containing internal cracks, with potential implications for enhancing the durability and sustainability of such structures. It is the first time to adopt the innovative machine learning algorithm with the combination of MEA and ABA-BPANN to forecast the gas permeability for sustainable cement mortar. Meanwhile, the traditional machine learning models of GA and PSO-optimised ABA-BPANN, MEA-optimised ELM and BPANN algorithms were constructed to validate the predicting performance of the MEA-tuned ABABPANN model. Moreover, utilising the validated MEA-tuned ABABPANN model as a foundation, a sensitivity analysis was conducted to quantitatively evaluate the relative significance of input parameters in determining the gas permeability of sustainable cracked cement mortar. Subsequently, an analytical formula was derived, enabling accurate estimation of gas permeability by individuals who may lack familiarity with the requisite skills.

The article suggests that the MEA-tuned ABABPANN machine learning model has demonstrated superior predictive performance compared to other conventional machine learning models. The article also implies that the new model has the potential to outperform existing methodologies in forecasting accuracy and efficiency. Following the sensitivity analysis results, it has been observed that the influence of internal crack characteristics, such as length, width, thickness, and quantity, on gas permeability of eco-friendly cement mortar with internal cracks, surpasses that of stress conditions, including confining pressure, pore pressure, and confining pressure loading/unloading phase.

The evaluation of gas permeability in sustainable cement mortar that contains internal cracks under various stress states requires consideration of multiple impact variables with intricate action mechanisms. This presents a daunting task for practitioners operating in this domain. The present study introduces an innovative machine-learning algorithm that can effectively address the challenge of accurately predicting the gas permeability of sustainable cement mortar, especially in the presence of internal cracks under different stress conditions. The successful implementation of this algorithm represents a significant step towards mitigating the inherent limitations and uncertainties associated with the design of sustainable cement mortar structures. In essence, the algorithm’s ability to provide precise and efficient gas permeability forecasts holds the key to enhancing the robustness and sustainability of cement mortar buildings.

## Figures and Tables

**Figure 1 materials-16-05330-f001:**
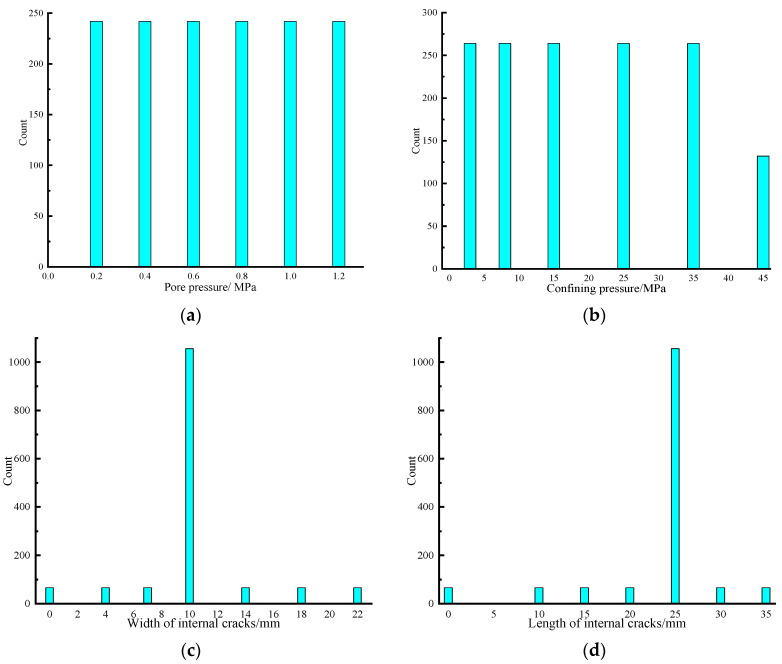
Data distributions of the built database. (**a**) Pore pressure; (**b**) Confining pressure; (**c**) Width of internal cracks; (**d**) Length of internal cracks; (**e**) Thickness of internal cracks; (**f**) Number of internal cracks; (**g**) Loading or unloading state.

**Figure 2 materials-16-05330-f002:**
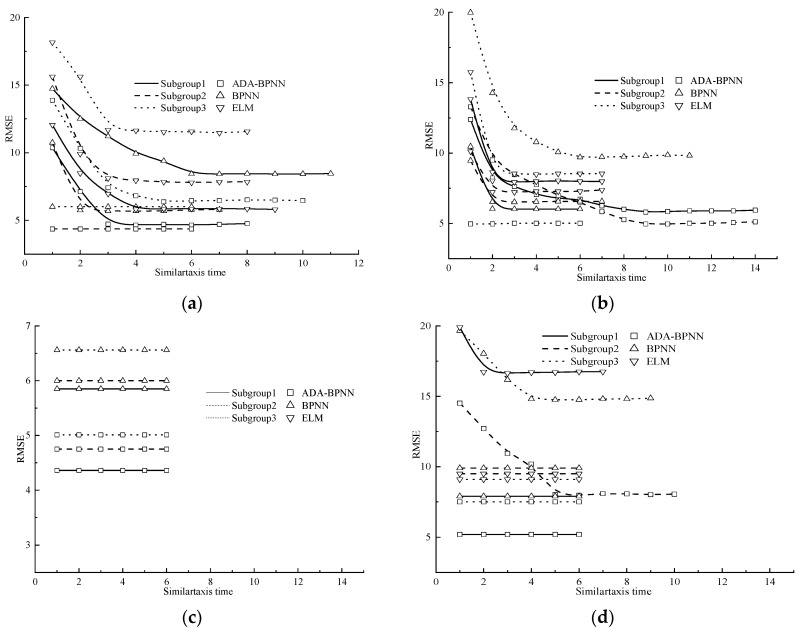
Revolution law of RMSE during the similartaxis operation. (**a**) Initial superior subgroups; (**b**) Initial temporary subgroups; (**c**) Superior subgroups after dissimilation operations; (**d**) Temporary subgroups after dissimilation operations.

**Figure 3 materials-16-05330-f003:**
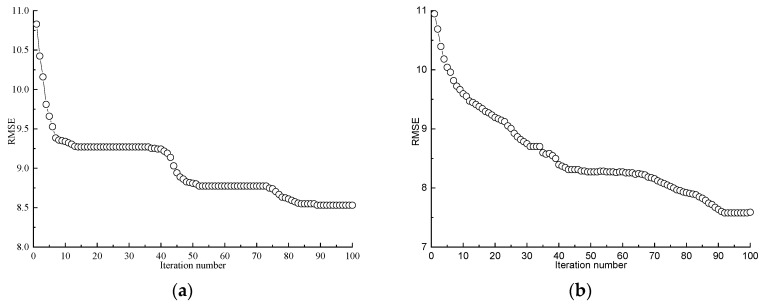
The optimising processes of the ABA-BPANN model based on GA and PSO. (**a**) GA-ABABPANN; (**b**) PSO-ABABPANN.

**Figure 4 materials-16-05330-f004:**
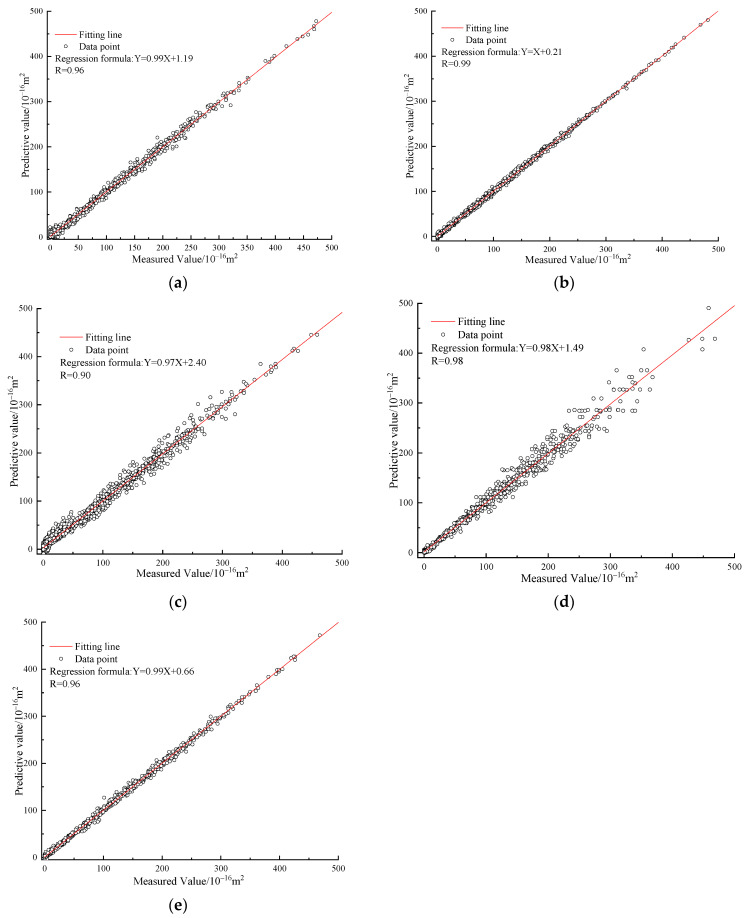
R values for training dataset. (**a**) MEA-BPANN; (**b**) MEA-ABABPANN; (**c**) MEA-ELM; (**d**) GA-ABABPANN; (**e**) PSO-ABABPANN.

**Figure 5 materials-16-05330-f005:**
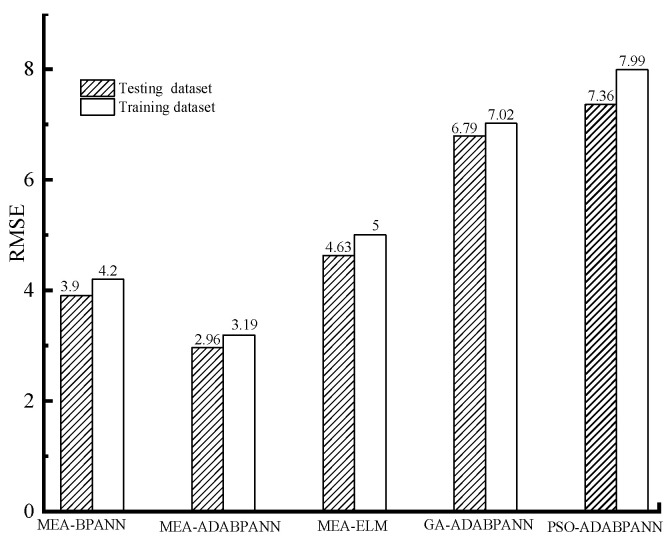
RMSE of the models.

**Figure 6 materials-16-05330-f006:**
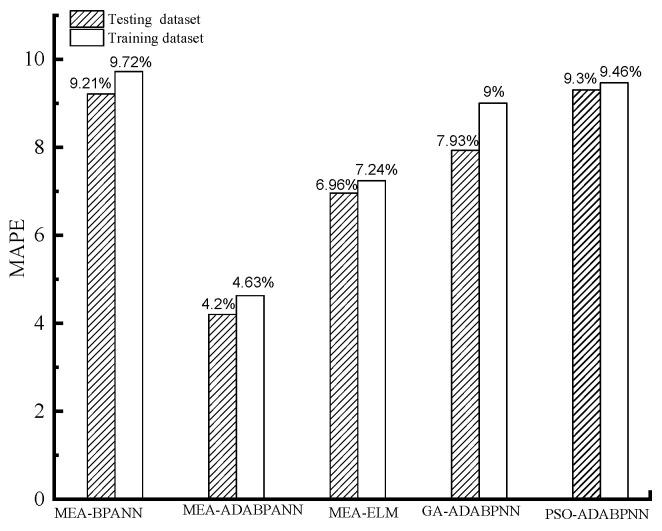
MAPE of the models.

**Figure 7 materials-16-05330-f007:**
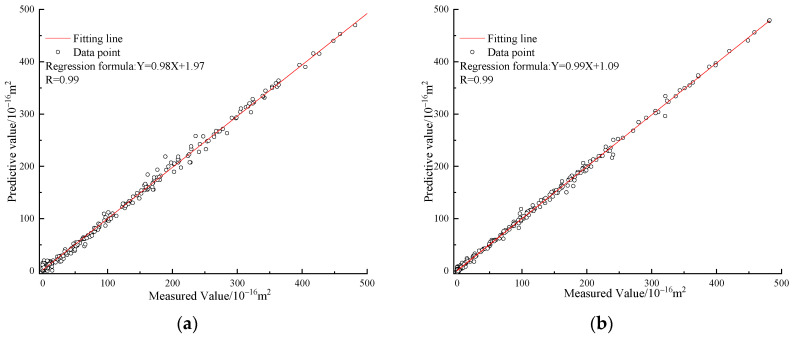
*R* values for testing dataset. (**a**) MEA-BPANN; (**b**) MEA-ABABPANN; (**c**) MEA-ELM; (**d**) GA-ABABPANN; (**e**) PSO-ABABPANN.

**Figure 8 materials-16-05330-f008:**
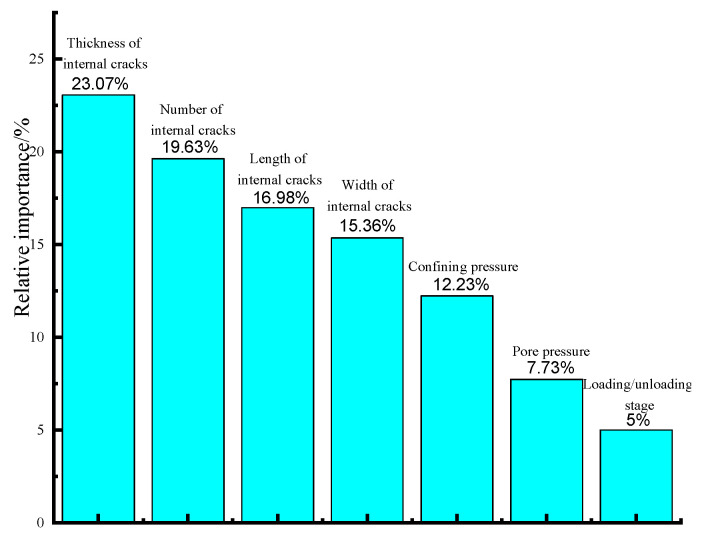
Relative importance of the input variables.

**Figure 9 materials-16-05330-f009:**
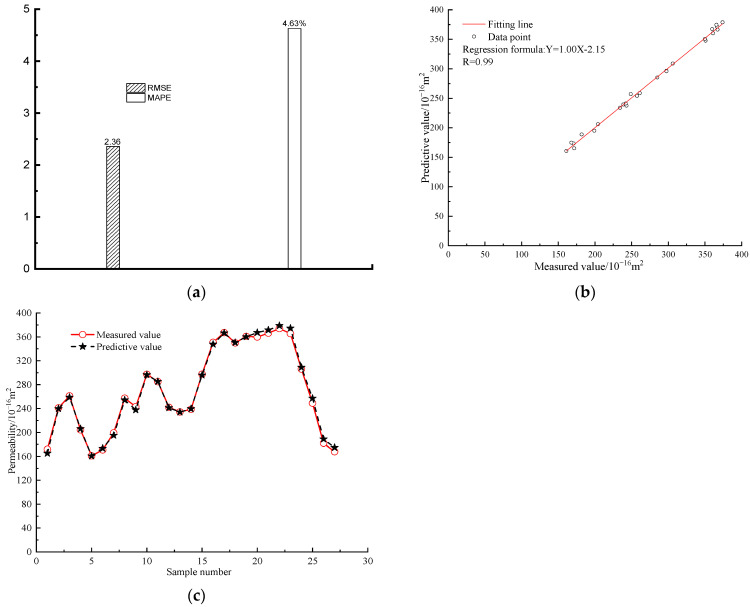
The validation results. (**a**) The MAPE and RMSE; (**b**) The *R* values; (**c**) The measured value and predictive value.

**Table 1 materials-16-05330-t001:** The specification of built algorithms.

Algorithms	Input Layer Joint Number	Hidden Layer Joint Number	Output Layer Joint Number	Base Learner Number
BPANN	7	9	1	X
ELM	7	27	1	X
ABABPANN	7	9	1	40

where, X means none.

**Table 2 materials-16-05330-t002:** The specification of the optimising algorithms.

Algorithms	MEA	PSO	GA
Number of Particles	600	200	200
Number of Subgroups	20		
Number of the Maximum Iterations	10	100	100
Particle numbers of the subgroup	3	O	O
Cognitive constant/Social constant	O	1.29	O
Inertia weight	O	0.79	O
Var maximum	O	2	2
Var minimum	O	−2	−2
Maximum velocity	O	1	O
Minimum velocity	O	0.02	O
Selection method	O	O	Roulette wheel
Crossover	O	O	Uniform
Mutation	O	O	Uniform
Mutation rate	O	O	0.24
Selection pressure	O	O	4

**Table 3 materials-16-05330-t003:** The optimised hyperparameters.

Algorithms	Hyperparameter	Optimising Magnitudes
BPANN	Original joint weight	−5–5
	Original joint threshold	−10–10
ELM	Original joint weight	2^−6^–2^6^
	The initial thresholds of joints	4^−6^–4^6^
ABA-BPANN	Base learner number	1–20
	Original joint weight in the base learner	−5–5
	Original joint threshold in the base learner	−10–10

**Table 4 materials-16-05330-t004:** Physical parameters of specimens.

Sample Number	Number of Internal Cracks (mm)	Thickness of Internal Cracks (mm)	Width of Internal Cracks (mm)	Length of Internal Cracks (mm)	Mass (g)
C-1	0	0	0	0	195.36
C-2	2	1.5	10	25	187.60
C-3	4	1.5	10	25	195.30
C-4	6	1.5	10	25	191.60
C-5	8	1.5	10	25	202.70
C-6	10	1.5	10	25	202.90
C-7	12	1.5	10	25	196.40
C-8	6	1.5	10	10	199
C-9	6	1.5	10	15	194
C-10	6	1.5	10	20	192
C-11	6	1.5	10	30	183
C-12	6	1.5	10	35	183
C-13	6	1.5	4	25	199
C-14	6	1.5	7	25	194
C-15	6	1.5	14	25	190
C-16	6	1.5	18	25	183
C-17	6	1.5	22	25	183
C-18	6	0.2	10	25	187.60
C-19	6	0.5	10	25	195.30
C-20	6	0.8	10	25	191.60
C-21	6	1.1	10	25	202.70
C-22	6	1.3	10	25	202.90

**Table 5 materials-16-05330-t005:** The mechanical properties of the sustainable cement mortar.

Property	Value
Density (g/cm^3^)	1.96
Uniaxial compressive strength (MPa)	61.3
Internal Cohesion (MPa)	11.1
Friction angle (°)	46.3
Poisson’s ratio	0.40
Elastic modulus (GPa)	11.9

**Table 6 materials-16-05330-t006:** Test plan.Experimental scheme.

Loading Confining Pressure (MPa)	Unloading Confining Pressure (MPa)	Pore Pressure (MPa)	Permeability Measuring Approach
3, 8, 15, 25, 35, 45	35, 25, 15, 8, 3	0.2, 0.4, 0.6, 0.8, 1.0, 1.2	Gas Flow Method

**Table 7 materials-16-05330-t007:** The statistics of the parameters.

Parameter	Categorise	Data Kind	Statistical Parameter/Magnitude			
			Minimum	Maximum	Average	Standard Deviation
Confining pressure (MPa)	Input parameter	Numeric	1	45	23	14
Pore pressure (MPa)		Numeric	0.2	1.2	0.7	0.1
Length of internal cracks (mm)		Numeric	0	35	17.5	2
Width of internal cracks (mm)		Numeric	0	22	11	1.4
Thickness of internal cracks (mm)		Numeric	0	1.5	0.75	0.11
Number of internal cracks		Numeric	0	12	6	1
Loading/unloading stage		Nominal	Loading and unloading state			
Gas permeability (10^−16^ m^2^)	Output parameter	Numeric	30.27	500.79	200.05	60.43

**Table 8 materials-16-05330-t008:** Mean joint weights and biases of the MEA-ABABPANN algorithm.

Hidden Layer Joint Number	Input Parameters							Weights	Biases	
								Output Parameter	Hidden Layer	Output Layer
	*Ps*	*Pc*	*S*	*L*	*W*	*N*	*T*	*k*		
1	0.23	0.30	0.14	−0.41	0.29	0.30	0.24	0.46	0.36	−0.27
2	0.93	0.36	0.30	0.46	0.36	−1.14	0.63	−2.32	−1.22	
3	−0.90	1.09	0.60	−1.46	0.46	−2.19	−1.19	−1.46	2.37	
4	−1.24	−0.17	0.70	2.02	1.22	3.33	−2.36	0.74	−0.42	
5	3.36	0.33	1.04	3.31	0.72	0.27	3.79	−3.36	1.19	
6	−0.90	1.04	−0.92	1.01	−0.45	−2.27	0.27	−3.72	2.46	
7	1.42	−1.02	−1.36	−2.14	1.46	1.36	0.64	2.66	3.36	
8	2.09	−2.45	2.17	−0.74	−1.42	−2.45	1.74	1.66	−0.72	
9	−0.20	−1.36	−3.33	0.79	0.24	0.72	−2.63	−0.45	0.63	

**Table 9 materials-16-05330-t009:** The basic parameters of prepared sample.

Sample Number	Number of Internal Cracks	Thickness of Internal Cracks (mm)	Width of Internal Cracks (mm)	Length of Internal Cracks (mm)
V-1	3	0.6	14	27
V-2	9	0.9	6	6
V-3	11	1.4	20	15

**Table 10 materials-16-05330-t010:** Test plan.

Loading Confining Pressure (MPa)	Pore Pressure (MPa)	Permeability MeasurementMethod
6, 20, 42	0.3, 0.6, 1.1	Gas Flow Method

## Data Availability

Not applicable.

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
