# Peer review of "Predicting the Gas Permeability of Sustainable Cement Mortar Containing Internal Cracks by Combining Physical Experiments and Hybrid Ensemble Artificial Intelligence Algorithms"

_materials, 2023, doi:10.3390/ma16155330_

Round 1

Reviewer 1 Report

Thank you for this contribution. This is an interesting and timely manuscript. This paper discusses how novel machine learning can be used in concrete materials. The conducted analysis is typically standard and falls within the expected work from such a publication and hence the work merits publication. As such, the authors are invited to properly address the following items:

1. In general, the introduction is light and does not represent state of the art in this domain. The amount of work in this area continues to rise rapidly. The authors are advised to strengthen their literature review section with supplementary material. Perhaps the addition of 1-2 pages can help strengthen this section. The following are some review papers on ML that can help identify relevant works (https://doi.org/10.1007/s11831-022-09793-w, https://doi.org/10.1016/j.jobe.2020.101827, https://doi.org/10.1016/j.eswa.2020.114060). In addition, the authors are invited to review the open literature for additional works.

2. What does this model tell us about the behavior of concrete that we did not know before? In other words, how do the new results match our physics and domain knowledge?

3. For a ML-based work, a question arises as to how our readers can benefit from the developed models. Thus, the authors are advised to consider providing their code and database for interested researchers to extend and benefit from this work. For example, the authors may option to upload this database into Mendeley or attach it to this paper. The same can be done for the code.

4. The presented equation seems complicated. Is there a way to simplify it?

5. How was the training process? Did the authors use cross validation? 

Author Response

Respond to Decision letter

From Zhiming Chao , Chuanxin Yang , Wenbing Zhang , Ye Zhang , Tianxing Zhou , Yutong Zhang  and Jiaxin Zhou

College of Ocean Science and Engineering,Shanghai Maritime University, Shanghai, China

Mentverse Ltd. 25 Cabot Square,Canary Wharf,London E14 4QZ,United Kingdom

To Editors

Journal: Materials

Re: Materials-2474619, title: " Predicting the gas permeability of sustainable cement mortar containing internal cracks by combining physical experiments and hybrid ensemble artificial intelligence algorithms "

Dear Reviewer,

Thank you for the Reviewer’s comments, guidance and suggestions on improving the manuscript. This manuscript has been modified according to the reviewer’ comments. In addition, we have improved the language quality of this manuscript. The modified parts in the revised paper have been highlighted with red colour.

In the following we address the reviewers’ comments one-by-one, by first referring to the comments and then stating the revisions we made accordingly, and/or providing a brief answer.

Comments and Suggestions for Authors

Thank you for this contribution. This is an interesting and timely manuscript. This paper discusses how novel machine learning can be used in concrete materials. The conducted analysis is typically standard and falls within the expected work from such a publication and hence the work merits publication. As such, the authors are invited to properly address the following items:

  1. In general, the introduction is light and does not represent state of the art in this domain. The amount of work in this area continues to rise rapidly. The authors are advised to strengthen their literature review section with supplementary material. Perhaps the addition of 1-2 pages can help strengthen this section. The following are some review papers on ML that can help identify relevant works (https://doi.org/10.1007/s11831-022-09793-w, https://doi.org/10.1016/j.jobe.2020.101827, https://doi.org/10.1016/j.eswa.2020.114060). In addition, the authors are invited to review the open literature for additional works.

Action 1: The author thanks the reviewer for their comments. We cite your recommended review papers (https://doi.org/10.1007/s11831-022-09793-w, https://doi.org/10.1016/j.jobe.2020.101827, https://doi.org/10.1016/j.eswa.2020.114060) on machine learning that help strengthen the literature review section. The corresponding content has been added into the modified manuscript, with being highlighted as red colour.

  1. What does this model tell us about the behavior of concrete that we did not know before? In other words, how do the new results match our physics and domain knowledge?

Action 2: The authors thank the reviewer for the comments. The purpose of this model is to save effort and time, and to make it easier to predict the air permeability of sustainable cement mortars with internal cracks under different stress environments. The value of gas permeability predicted from the proposed model is approximate to the permeability obtained from the physical test

  1. For a ML-based work, a question arises as to how our readers can benefit from the developed models. Thus, the authors are advised to consider providing their code and database for interested researchers to extend and benefit from this work. For example, the authors may option to upload this database into Mendeley or attach it to this paper. The same can be done for the code.

Action 3: The authors thank the reviewer for the comments. We will upload this database and code into Mendeley.

  1. The presented equation seems complicated. Is there a way to simplify it?

Action 4: The authors thank the reviewer for the comments. We have simplified the presented equation(16-25) in the revised manuscript.

  1. How was the training process? Did the authors use cross validation?

Action 5: The authors thank the reviewer for the comments. In the training process, the fitness function value (RMSE) of individuals was computed via using k-fold cross-validation method (k-CV) based on the training dataset of the built database, with k being taken as 10 according to the size of database and the recommendations in existing investigation[1,2].

References

[1] Witten IH, Frank E, Hall MA, Pal CJ. Data Mining: Practical machine learning tools and techniques: Morgan Kaufmann; 2016.

[2] Rodriguez JD, Perez A, Lozano JA. Sensitivity analysis of k-fold cross validation in prediction error estimation. IEEE transactions on pattern analysis and machine intelligence. 2009;32:569-575.

The above responses should answer all the questions. If you have any more questions, please let us know.

Sincerely Yours,

Zhiming Chao , Chuanxin Yang , Wenbing Zhang , Ye Zhang , Tianxing Zhou , Yutong Zhang  and Jiaxin Zhou

Reviewer 2 Report

The authors have done a large amount of research. The manuscript contains a enough amount of graphic materials and tables and is of scientific interest. However, the presentation of the material in a number of sections of the manuscript requires significant adjustments and clarifications.

1. It is recommended to add the following words to the list of keywords: gas permeability and Mind Evolutionary Algorithm

2. The abbreviation ADA-BPANN does not match the wording shown Adaptive Boosting Algorithm-Back Propagation Artificial Neural Network. Probably, instead of ADA there should be ABA. Please clarify.

3. The authors talk about the need to study the gas permeability of concrete made from industrial waste, but this is also true for concrete based on traditional raw materials. Recommended to reconsider the introduction and clearly justify the choice of the object of study.

4. Section 4 "Physical test". The authors indicate that discarded concrete (L 226) was used as the infill, but further down the text it is indicated that sand (L233) was used as the infill. It is recommended that this paragraph be revised (L 225-235). Clearly indicate the raw materials from which the samples were made and indicate the characteristics of these materials.

5. Table 4. It is not clear what the difference between the compositions. Previously, the authors indicated that the samples were prepared at the same ratio of components (L233), then why the difference (by weight) between the number of samples (for example, C-3 and C-11) is 20 g (about 10%). Please clarify this.

6. It is not clear which properties of which particular composition are given in Table 5. Please clarify what the authors understand by the term "sustainable cement"; 7. Table 6. Why are 6 values of Loading confining pressure (MPa) specified, while Unloading confining pressure (MPa) has 5 values. The value "45 MPa" (Unloading confining pressure (MPa)) is probably missing. Please clarify this.

Author Response

Respond to Decision letter

From Zhiming Chao , Chuanxin Yang , Wenbing Zhang , Ye Zhang , Tianxing Zhou , Yutong Zhang  and Jiaxin Zhou

College of Ocean Science and Engineering,Shanghai Maritime University, Shanghai, China

Mentverse Ltd. 25 Cabot Square,Canary Wharf,London E14 4QZ,United Kingdom

To Editors

Journal: Materials

Re: Materials-2474619, title: " Predicting the gas permeability of sustainable cement mortar containing internal cracks by combining physical experiments and hybrid ensemble artificial intelligence algorithms "

Dear Reviewer,

Thank you for the Reviewer’s comments, guidance and suggestions on improving the manuscript. This manuscript has been modified according to the reviewer’ comments. In addition, we have improved the language quality of this manuscript. The modified parts in the revised paper have been highlighted with red colour.

In the following we address the reviewers’ comments one-by-one, by first referring to the comments and then stating the revisions we made accordingly, and/or providing a brief answer.

Comments and Suggestions for Authors

The authors have done a large amount of research. The manuscript contains a enough amount of graphic materials and tables and is of scientific interest. However, the presentation of the material in a number of sections of the manuscript requires significant adjustments and clarifications.

  1. It is recommended to add the following words to the list of keywords: gas permeability and Mind Evolutionary Algorithm

Action 1: The authors thank the reviewer for the comments. We have added gas permeability and Mind Evolutionary Algorithm to the list of keywords in the revised manuscript.

  1. The abbreviation ADA-BPANN does not match the wording shown Adaptive Boosting Algorithm-Back Propagation Artificial Neural Network. Probably, instead of ADA there should be ABA. Please clarify.

Action 2: The authors thank the reviewer for the comments. We have changed the abbreviation ADA to ABA in the modified manuscript.

  1. The authors talk about the need to study the gas permeability of concrete made from industrial waste, but this is also true for concrete based on traditional raw materials. Recommended to reconsider the introduction and clearly justify the choice of the object of study.

Action 3: The authors thank the reviewer for the comments. We have reconsidered the introduction and clearly justify the choice of the object of study. The corresponding content has been added into the modified manuscript, with being highlighted as red colour.

  1. Section 4 "Physical test". The authors indicate that discarded concrete (L 226) was used as the infill, but further down the text it is indicated that sand (L233) was used as the infill. It is recommended that this paragraph be revised (L 225-235). Clearly indicate the raw materials from which the samples were made and indicate the characteristics of these materials.

Action 4: The authors thank the reviewer for the comments. We have indicated the raw materials from which the samples were made and indicated the characteristics of these materials. The corresponding content has been added into the modified manuscript, with being highlighted as red colour.

  1. Table 4. It is not clear what the difference between the compositions. Previously, the authors indicated that the samples were prepared at the same ratio of components (L233), then why the difference (by weight) between the number of samples (for example, C-3 and C-11) is 20 g (about 10%). Please clarify this.

Action 5: The authors thank the reviewer for the comments.The samples were prepared at the same ratio of components and volume, but the number, thickness, width and length of internal cracks are different, resulting in different porosities or densities, so the weight between the number of samples is different.

  1. It is not clear which properties of which particular composition are given in Table 5. Please clarify what the authors understand by the term "sustainable cement"; 7. Table 6. Why are 6 values of Loading confining pressure (MPa) specified, while Unloading confining pressure (MPa) has 5 values. The value "45 MPa" (Unloading confining pressure (MPa)) is probably missing. Please clarify this.

Action 6: The authors thank the reviewer for the comments. In this test, the replacement ratio of sustainable aggregates is 100 %, and all the aggregate of sustainable cement is derived from the discarded concrete. The cement mortar is generated according to a mass ratio and then mashed, and the cement mortar after mashing as discarded concrete was used as the infill. The 45MPa is the maximum loading confining pressure, and the unloading confining pressure can only be less than the maximum loading confining pressure, so 45MPa can not be used as the unloading confining pressure.

The above responses should answer all the questions. If you have any more questions, please let us know.

Sincerely Yours,

Zhiming Chao , Chuanxin Yang , Wenbing Zhang , Ye Zhang , Tianxing Zhou , Yutong Zhang  and Jiaxin Zhou

Reviewer 3 Report

In this research, a comprehensive database consisting of a large number of gas permeability tests was assembled.  

- In the introduction,  is a large amount of basic knowledge. I suggest that the authors shorten this part by focusing on the degree of novelty that they bring in comparison with the methods described in the literature.

- Detailing the procedures for carrying out the tests in Table 4 could lead to an easier understanding of the experiment. How was the number of cracks evaluated? What were the conditions for keeping the evidence before the evaluation?

- What is the standard (ISO) by which the gas permeability has been evaluated?

Author Response

Respond to Decision letter

From Zhiming Chao , Chuanxin Yang , Wenbing Zhang , Ye Zhang , Tianxing Zhou , Yutong Zhang  and Jiaxin Zhou

College of Ocean Science and Engineering,Shanghai Maritime University, Shanghai, China

Mentverse Ltd. 25 Cabot Square,Canary Wharf,London E14 4QZ,United Kingdom

To Editors

Journal: Materials

Re: Materials-2474619, title: " Predicting the gas permeability of sustainable cement mortar containing internal cracks by combining physical experiments and hybrid ensemble artificial intelligence algorithms "

Dear Reviewer,

Thank you for the Reviewer’s comments, guidance and suggestions on improving the manuscript. This manuscript has been modified according to the reviewer’ comments. In addition, we have improved the language quality of this manuscript. The modified parts in the revised paper have been highlighted with red colour.

In the following we address the reviewers’ comments one-by-one, by first referring to the comments and then stating the revisions we made accordingly, and/or providing a brief answer.

Comments and Suggestions for Authors

In this research, a comprehensive database consisting of a large number of gas permeability tests was assembled. 

1、In the introduction,  is a large amount of basic knowledge. I suggest that the authors shorten this part by focusing on the degree of novelty that they bring in comparison with the methods described in the literature.

Action 1: The authors thank the reviewer for the comments. We have shortened this part by focusing on the degree of novelty in the modified manuscript.

2、Detailing the procedures for carrying out the tests in Table 4 could lead to an easier understanding of the experiment. How was the number of cracks evaluated? What were the conditions for keeping the evidence before the evaluation?

Action 2: The authors thank the reviewer for the comments. The number of cracks is prefabricated according to different test conditions. An alternative method was adopting cement mortaras rock-like material and inserting tin sheets into mold before pouring mortar[1]. In the low-temperature environment, the tin sheets in the cement mortar become loose non-strength tin power, forming smooth hidden cracks in the cement mortar specimen. The dimension of the smooth hidden cracks is the same with the solid tin sheets, which can be adjusted by changing the dimension of the solid tin sheets. In order to validate the effectiveness of the proposed method, the jointed specimens with different thicknesses of internal cracks were selected randomly from the prepared jointed specimens. The selected jointed specimens were cut open transversely from the central axis of the sample by using the rock cutting machine. The hidden joints in the selected specimens were numbered and measured. the prepared jointed specimens were extremely close to the required dimension, and the deviation between the measured dimension and the required dimension could be negligible. Hence, the prepared jointed specimens can be used to conduct the physical model experiments.

3、What is the standard (ISO) by which the gas permeability has been evaluated?

Action 3: The authors thank the reviewer for the comments. Gas Flow Method was adopted in this experiment to measure permeability of the specimens. Gas Flow Method is based on the assessment of the mean gasflow that penetrates through specimen to calculate the permeability. This method has been used for many years, and many papers have been published on its application[2-5].

References

[1]Chao, Z.; Ma, G.; Hu, X.; He, K.; Luo, G.; Mei, X.; Li, D., Experimental research on stress-dependent permeability and porosity of rock-like materials with different thicknesses of smooth hidden joints. Int. J. Mod. Phys. B. 2020, 34, (12), 2050117.

[2]Tanikawa, W., & Shimamoto, T. (2009). Comparison of Klinkenberg-corrected gas permeability and water permeability in sedimentary rocks. International Journal of Rock Mechanics and Mining Sciences, 46(2), 229-238.

[3]Chao, Z., Ma, G., Hu, X., Luo, G., Experimental research on stress-dependent permeability and porosity of compact sand-stone with different moisture saturations. J.Nat.Gas.Sci.Eng. 2020, 84, 103639.

[4]Chao, Z., Ma, G., Wang, M., Experimental and numerical modelling of the mechanical behaviour of low-permeability sand-stone considering hydromechanics. Mech. Mater. 2020, 148, 103454.

[5]Chao, Z., Gong, B., Yue, W., Xu, X., Shi, D., Yang, C., Hu, T., Experimental study on stress-dependent gas permeability and porosity of artificially cracked cement mortar. Constr. Build. Mater. 2022, 359, 129290.

The above responses should answer all the questions. If you have any more questions, please let us know.

Sincerely Yours,

Zhiming Chao , Chuanxin Yang , Wenbing Zhang , Ye Zhang , Tianxing Zhou , Yutong Zhang  and Jiaxin Zhou

Round 2

Reviewer 1 Report

.

.

Reviewer 2 Report

Accept in present form

Reviewer 3 Report

The authors responded to all my comments. 

If the editor agrees can be published in the present form!